# Pilot Study on Poultry Meat from Antibiotic Free and Conventional Farms: Can Metagenomics Detect Any Difference?

**DOI:** 10.3390/foods11030249

**Published:** 2022-01-18

**Authors:** Alessandra De Cesare, Chiara Oliveri, Alex Lucchi, Federica Savini, Gerardo Manfreda, Claudia Sala

**Affiliations:** 1Department of Veterinary Medical Sciences, University of Bologna, Ozzano dell’Emilia, 40064 Bologna, Italy; federica.savini3@unibo.it; 2Department of Agricultural and Food Sciences, University of Bologna, Ozzano dell’Emilia, 40064 Bologna, Italy; chiara.oliveri5@unibo.it (C.O.); alex.lucchi3@unibo.it (A.L.); gerardo.manfreda@unibo.it (G.M.); 3Department of Physics and Astronomy, University of Bologna, 40126 Bologna, Italy; claudia.sala3@unibo.it

**Keywords:** broiler, antibiotic free farms, conventional farms, microbiome, resistome, shotgun metagenomic sequencing

## Abstract

Antibiotic free farms are increasing in the poultry sector in order to address new EU regulations and consumer concerns. In this pilot study, we investigated whether the efforts of raising chickens without the use antibiotics make any difference in the microbiome of poultry meat eaten by consumers. To this aim we compared the microbiomes characterizing caeca and the corresponding carcasses of two groups of chickens reared, one reared on a conventional farm and one on an antibiotic-free intensive farm. The results showed a clear separation between the taxonomic, functional and antibiotic resistant genes in the caeca of the birds reared on the conventional and antibiotic free farm. However, that separation was completely lost on carcasses belonging to the two groups. The antibiotic-free production resulted in statistically significant lower antimicrobial resistance load in the caeca in comparison to the conventional production. Moreover, the antimicrobial resistance load on carcasses was much higher than in the caeca, without any significant difference between carcasses coming from the two types of farms. All in all, the results of this research highlighted the need to reduce sources of microbial contamination and antimicrobial resistance not only at the farm level but also at the post-harvest one.

## 1. Introduction

In 2019 the European Union (EU) produced an estimated 13.3 million tonnes of poultry meat, representing an increase of around 27% in comparison to 2010 [1]. Poultry meat is characterized by high-quality proteins, vitamins, and minerals important for the human diet [2]. Since the poultry rearing cycle lasts 35 to 42 days, poultry meat can be produced without the use of antimicrobials more easily than pork and beef, which have rearing cycles that last for months. Moreover, the mean values, expressed in number of defined daily doses (DDDvet)/biomass for poultry, of antimicrobial agents obtained from the technical estimates of the sales of veterinary antimicrobials in the European Union in 2016 were 0.5 for poultry versus 1.3 for pigs [3].

There are different strategies to achieve antibiotic free poultry flocks such as the implementation of effective biosafety measures and management options as well as the promotion of beneficial microbes in the gastrointestinal (GI) tract of chickens to enhance animal health and inhibit pathogen colonization. To this aim, feed can be supplemented with probiotics and prebiotics, and also blended in the same supplement (i.e., a symbiotic), to ensure diversity and stability of the GI microbial community, as well as positive interactions with the host’s gastroenteric epithelium and immune system [4]. As an alternative, probiotics can also be supplemented in the litter and up taken by the animals [5]. Whenever a poultry disease occurs in an antibiotic free flock the animals are treated with antibiotics and the flock turns into conventional. Therefore, poultry farms can occasionally rear antibiotic free flocks along with conventional ones.

In the last decades, the microorganisms colonizing the chicken gut have been mainly investigated using classical microbiological cultural methods. However, the application of high-throughput sequencing techniques is now allowing a deeper investigation of the whole microbial composition of the chicken GI tract, including strictly anaerobic and not cultivable microorganisms. With reference to the high-throughput sequencing techniques, Durazzi et al. [6] compared taxonomic results obtained by metataxonomics (i.e., 16S rRNA gene sequencing) and metagenomics (i.e., whole shotgun metagenomic sequencing) to investigate their reliability for bacteria profiling of the chicken GI tract. The results showed that shotgun sequencing has more power to identify less abundant taxa than 16S sequencing, and that the low abundant taxa have a relevant biological meaning, being able to discriminate between the experimental conditions as much as the more abundant genera detected by both sequencing strategies. Moreover, shotgun metagenomics allows for the investigation not only of the microbial population, but also of the composition in functional genes, including antibiotic resistant ones [7,8,9].

The increasing incidence of antibiotic resistance is considered one of the major threats to global public health [10,11]. In the European Union, antimicrobial resistance is responsible for 33,000 human deaths per year [12]. The use of antibiotics in animal food productions has been considered the primary cause of shedding of antibiotic resistant pathogenic and commensal bacteria [13], which can then be transferred to humans via several pathways, ref. [14] including the food-chains [15,16]. Tang et al. [17] demonstrated that reducing the level of antibiotic use in animal productions can be effective to fight antibiotic resistance in both animals and humans. To this aim, many initiatives and strategies have been put in place by both policy makers and producers and as a possible consequence, in 2017, for the first time since 2011, the antimicrobial consumption in food producing animals was lower than in humans (i.e., 108.3 mg/kg vs. 130 mg/kg) [3].

To limit the use of antibiotics in animal rearing, Regulation (EC) No. 1831/2003 [18] banned the marketing and use of antibiotics as growth promoters in feed in the European Union since January 2006. Furthermore, starting from January 2022, the new Regulations EU 2019/6 [19] on Veterinary Medicinal Products and 2019/4 [20] on Medicated Feed will enter in force as further steps in the EU strategy to fight antimicrobial resistance. Other mitigation measures include the prevention of bacterial diseases in the animals, the use of specific antimicrobials for food producing animals but not for humans, the improvement of biosecurity protocols at farm level, the wide use of vaccination programs, and the use of nutrients and additives in feed promoting the immune system and supporting beneficial microbes [21].

In the pilot study described in this paper we investigated whether the efforts of raising chickens without the use of antibiotics made any difference in the microbiome of poultry that consumers eat. To this end, we compared the microbiomes characterizing caeca and the corresponding carcasses of two groups of chickens reared on different farms—one conventional and one antibiotic-free. Moreover, with the view of planning future studies, we investigated whether checking the correlation between the microbiome and resistome in the caeca and the carcass of the same animal provides more insight than the same analysis performed at flock level.

## 2. Materials and Methods

### 2.1. Samples and Sampling Plan

A total of 30 slaughtered broilers were randomly sampled at the slaughterhouse within the first group of animals processed at the start of the working day, when the slaughterhouse was still clean and disinfected to avoid bias due to cross-contamination. Among the selected broilers, 15 belonged to a poultry flock reared in a conventional farm and treated with Amoxicillin (20–30 mg/kg of live weight in drinking water for three to five days) at days eight and 29 and with sulfadimethoxine/trimethoprim (100 mg/20 mg in 1–2 L of drinking water once a day for five days) at day 24. The remaining 15 belonged to a poultry flock reared in a conventional farm but never treated with antibiotics. All tested animals were female, ROSS, fed with no medicated feed and slaughtered at 47 (antibiotic free) and 48 (conventional) days. For each sampled animal, both the gastrointestinal tract and the carcass were collected. The GI tract was sampled during slaughtering, at the evisceration step, and immediately stored in a sterile plastic bag kept at 4 °C in a refrigerated box. Moreover, a sterile plastic flag was attached to the hook transporting the carcass from which the GI tract was sampled in order to pick up that specific carcass at the end of the refrigeration tunnel. After sampling, each carcass was stored in a sterile plastic bag kept at 4 °C in a second refrigerated box, different from that containing the GI tracts. All samples were transported to the laboratory within two hours and immediately processed.

Each GI tract was dissected and a small portion (i.e., 0.5 to 2 g) of caeca content was collected, transferred into a 2 mL sterile plastic tube, and flash frozen in liquid nitrogen before storage at −80 °C until DNA extraction. A total of 10 g of neck and breast skin were collected from each carcass, diluted in 90 mL of sterile physiological solution (NaCl 0.90%), homogenized in the stomacher (MAYO HG 400V, Italy) at normal speed for 1 min and centrifuged at 4 °C for 20 min at 9980× *g*. The obtained pellet containing the concentrated cells was stored at −80 °C until DNA extraction.

### 2.2. DNA Extraction and Sequencing

The DNA was extracted using a bead-beating procedure followed by the QIAmp^®^ DNA Stool Mini Kit (Qiagen, Milan, Italy) for samples of caeca content [22], and by PowerFood^®^ Microbial DNA Isolation Kit (MO BIO-Qiagen) for the pellet obtained from each carcass [23]. The DNA extracted from each sample was quantified using a BioSpectrometer^®^ (Eppendorf, Milan, Italy) to assess DNA yield, in terms of quantity and quality. Total DNA from caeca and carcass samples was fragmented and tagged with sequencing indexes and adapters using the Nextera XT DNA Library Preparation Kit (Illumina, San Diego, CA, USA). Shotgun metagenomic sequencing was performed using the NextSeq500 (Illumina) 2 × 150 bp in paired-end mode. One caeca sample was removed later in the process for technical reasons (linked to sequencing yield) resulting in a total of 59 metagenomes: 29 from broilers caeca and 30 from carcasses.

### 2.3. Bioinformatic and Biostatistic Analysis

Filtering and trimming of raw reads were performed using MG-RAST https://www.mg-rast.org (accessed on 14 December 2021) [24] bioinformatics pipelines. In the taxonomic analysis, only taxa from the bacterial domain were considered. Moreover, taxa present in less than 4 samples or represented by less than four reads were discarded. Analogously, in the functional analysis, functional genes present in less than four samples or represented by less than 4 reads were discarded.

The statistical analysis of both the taxonomic and functional gene composition was performed using R 3.6.3 and the libraries phyloseq 1.30.0 [25] and DESeq2 1.26.0 [26]. Relative abundances displayed in the bar plots were computed normalizing to sum to 1 the read counts obtained from MG-RAST. Then, in the bar plots only the first (at most) 20 taxa/functional elements with relative abundance greater than 1% were shown.

Before proceeding with the statistical analysis, read counts were normalized with DESeq2 to take into account the different sizes of the samples. In this step, size factors were estimated using the function estimateSizeFactors of DESeq2 with the “poscount” option, as suggested when dealing with sparse data. Then, DESeq2 was used to assess whether the taxa/function abundances differed between groups. Specifically, the Wald test was used to determine the statistical significance and the Log Fold Changes were shrunk using the apeglm method [27]. Finally, p-values were adjusted for multiple testing using the Benjamini-Hocheberg procedure [28]. A threshold of 0.05 was used in all analyses to assess their statistical significance.

Alpha diversity was estimated using the InvSimpson/Shannon/Chao1 index, and differences in alpha diversity between groups were evaluated fitting a linear regression model and using the Student’s *t*-test to assess whether the linear relationship between alpha diversity and the grouping was negligible. Beta diversity was estimated starting from the read counts normalized with DESeq2 and computing the Bray-Curtis dissimilarity [29] among samples. Principal Coordinate Analysis (PCoA) was used to visualize the results. After applying the rlog transformation [26] to DESeq2 normalized counts, Principal Component Analysis (PCA) was performed using the function prcomp of the library stats 3.6.3 in R and the correlation between samples was computed using Kendall’s coefficient.

The resistome of each sample was predicted using the Resistance Gene Identifier (RGI) [30]. Fastq reads were aligned using the bowtie2 algorithm [31] to the ‘canonical’ curated CARD reference sequences [30], as well as to the in silico predicted allelic variants available in CARD’s Resistomes & Variants data set [30], as suggested in the resistance gene identifier (RGI). The alignments were obtained at the allele level and were filtered so that only entries with >95% identity to the CARD reference sequences and with more than 50 base pairs of reference allele covered by reads were kept. RGI mapping counts were adjusted for differences in both gene lengths and bacterial sequence abundances by computing fragments per kilobase reference per million bacterial fragments (FPKM). Results at the AMR gene family and drug class level were obtained by aggregating the counts at the allele level. The beta-diversity of the samples based on the resistome was obtained by computing the PCoA. To this aim, the counts were normalized with DESeq2 as previously described, and the Bray-Curtis distances between all samples were calculated using the R packages vegan 2.5.7 [32] and phyloseq 1.28.0 [25]. The PCoA was computed separately for caeca and carcass samples, and the effect of the origin of the sample on the sample dissimilarities were determined using permutational multivariate analysis of variance using distance matrices (the ‘adonis2′ function in the vegan v2.5.7 package). Finally, conventional and antibiotic-free AMR gene families were compared using the same DESeq2 pipeline previously described for the taxonomic and functional analysis.

## 3. Results

Metagenomic sequencing yielded an average of 5 Gbp/sample. The 59 metagenomes sequenced are publicly available from MG RAST at https://www.mg-rast.org/mgmain.html?mgpage=project&project=mgp89213 (accessed on 14 December 2021) and described in Appendix A.

### 3.1. Taxonomic and Functional Gene Composition of Caeca

The beta diversity index (generated using the Bray-Curtis distance metric) calculated at the genus level highlighted a clear dissimilarity in the community composition in the caeca sampled in the two groups of broilers tested (adonis2 *p*-value < 0.00001) (Figure 1).

Sixteen of the top 20 most abundant genera identified within the caeca were shared between the two tested groups (Figure 2). Moreover, *Alkaliphilus*, *Desulfibacterium*, *Bacillus* and *Ethanoligenens* were only detected in the caeca of birds from the conventional farm (Figure 2a), while *Coprococcus*, *Escherichia*, *Parabacteroides* and *Provotella* were detected in the caeca of birds from the antibiotic-free farm (Figure 2b).

The normalized mean counts of *Alkaliphilus*, *Bacillus*, *Desulfitobacterium*, *Ethanoligenens* and *Streptococcus* were significantly higher in the caeca of birds reared on the conventional farm, while those of *Alistipes*, *Anaerotruncus*, *Bacteroides*, *Coprococcus*, *Dorea*, *Escherichia*, *Holdemania*, *Lactobacillus*, *Parabacteroides*, *Prevotella*, *Roseburia*, *Ruminococcus* and *Subdoligranulum* were higher in the caeca of birds reared on the antibiotic-free farm (Appendix A). In terms of alpha diversity, representing richness and diversity of the genera within the caeca, the InvSimpson, Shannon and Chao1 indexes were significantly higher in the caeca of birds from the group of animals reared in the conventional in comparison to the antibiotic free farm (Appendix A and Figure 3).

The beta diversity index calculated for the functional genes confirmed a clear association of the caeca with their farm of origin (adonis2 *p*-value < 0.00001) (Figure 4).

Fourteen of the top 20 most abundant functional genes identified within the caeca of birds were shared between the two tested groups (Figure 5). Moreover, functional genes type I restriction−modification system-restriction subunit R (EC 3.1.21.3), site−specific recombinase, DNA topoisomerase III (EC 5.99.1.2), ferrous iron transport protein B, copper−translocating P−type ATPase (EC 3.6.3.4) and DNA gyrase subunit A (EC 5.99.1.3) were listed among the most abundant functional genes in the caeca of birds from the conventional farm, while chaperone protein DnaK, glutamine synthetase type III, GlnN (EC 6.3.1.2), leucyl−tRNA synthetase (EC 6.1.1.4), valyl−tRNA synthetase (EC 6.1.1.9), clpB protein and tonB−dependent receptor were only detected in the caeca of birds from the antibiotic-free farm.

Besides these qualitative differences, the normalized mean values of beta-galactosidase (EC 3.2.1.23) and DNA topoisomerase III EC 5.99.1.2 (in PFGI-1-like cluster) were significantly higher in the caeca of birds from the conventional farm, while those of integrase, translation elongation factor G, ribonucleotide reductase of class III (anaerobic)-large subunit EC 1.17.4.2, excinuclease ABC subunit A paralog were greater in the Bacteroides group, and the chaperone protein DnaK was significantly higher in the caeca of birds reared in the antibiotic free farm (Appendix A).

### 3.2. Taxonomic and Functional Composition of Carcasses

The genera identified in the carcasses showed a slight clustering (adonis2 *p*-value = 0.020) according to the farm of origin (Appendix A). Overall, 14 of the top 20 most abundant genera identified in the carcasses were shared between the two tested groups (Figure 6). Moreover, *Anoxybacillus*, *Bacillus*, *Flavobacterium*, *Pedobacter*, *Geobacillus* and *Sphingobacterium* were only detected on carcasses of birds from the conventional farm (Figure 6a), while *Aeromonas*, *Burkholderia*, *Endoriftia*, *Prevotella*, *Ruminococcus* and *Shewanella* were only detected on carcasses of birds from the antibiotic-free farm (Figure 6b).

The normalized mean values of Anoxybacillus, Bacillus, Geobacillus, Pedobacter, and Sphingobacterium were significantly higher on carcasses of birds reared in the conventional farm, while those of Aeromonas, Bacteroides, Prevotella and Ruminococcus were higher in the carcasses of birds reared on the antibiotic-free farm (Appendix A). The alpha diversity indexes (i.e., InvSimpson, Shannon and Chao1) calculated for bacteria genera identified on carcasses from the conventional and the antibiotic-free farm did not show any significative difference (Appendix A). Moreover, the beta diversity calculated for the functional genes did not group the carcasses according to the farm of origin (adonis2 *p*-value = 0.332) (Appendix A).

Three of the top 20 most abundant functional genes identified on the carcasses were shared between the two tested groups (Figure 7). Moreover, the functional genes cytochrome c oxidase polypeptide II (EC 1.9.3.1) and DNA-directed RNA polymerase beta’ subunit (EC 2.7.7.6) were listed among the most abundant functional genes on carcasses of birds from the conventional farm, while 2-oxoglutarate dehydrogenase E1 component (EC 1.2.4.2), RNA helicase putative and DNA topoisomerase I (EC 5.99.1.2) were most abundant only on carcasses of birds from the antibiotic-free farm.

As for the genera, the functional genes did not show significative different normalized mean counts between the two groups of carcasses tested (*p* < 0.05).

### 3.3. Longitudinal Analysis of Caeca and Carcass Microbiomes Belonging to the Same Animal

In Figure 8 the correlations between caeca samples and carcasses are shown. Both the dimension of the circle and the color scale represent the value of the correlation coefficient. Overall, a positive correlation was always detected between the genera colonizing the caeca and the corresponding carcass. However, such correlation was not stronger for samples collected from the same animal than for animals belonging to the same flock. As an example, bacteria genera identified in the caeca of the bird labelled as 31 in Figure 8a are well correlated to those of carcass 31 as well as to carcasses 36, 37, 43 and 45. On the contrary, the bacteria genera identified in the caeca of the bird labelled as 50 in Figure 8b show a higher correlation with the carcasses 47 and 59 in comparison to the carcass of the same animal (i.e., carcass 50). A positive correlation was also identified between functional genes detected in the caeca and corresponding carcass for chickens reared in the conventional (Figure 8c) and antibiotic free (Figure 8d) farms. For the functional genes, such positive correlation was higher in comparison to that observed for the bacteria genera, thus resulting in stronger blue dots.

### 3.4. Identification of Antibiotic Resistance Genes in the Caeca and Carcass Microbiome

The antibiotic resistance genes (ARGs) were retrieved from functional genes identified in the caeca and in the carcasses of the tested animals and classified as such in level 2 of the biological-function ontology in the SEED category [33]. Among the ARGs with normalized mean values of abundance >1000 in at least one tested group, the regulatory sensor-transducer-BlaR1/MecR1 family, UDP-N-acetylmuramoylalanyl-D-glutamate-2,6-diaminopimelate ligase (EC 6.3.2.13), macrolide export ATP-binding/permease protein MacB (EC 3.6.3.-), multi antimicrobial extrusion protein (Na(+)/drug antiporter)-MATE family of MDR efflux pumps, topoisomerase IV subunit B (EC 5.99.1.-) and vancomycin response regulator VanR were significantly higher in the caeca of birds reared in the conventional farm in comparison to the antibiotic-free farm (Appendix A). On the contrary, the acriflavin resistance protein was significantly higher in the caeca of birds reared on the antibiotic-free farm (Appendix A). As far as the carcasses are concerned, no differences were identified between normalized mean values of abundance of ARGs detected on carcasses sampled in the two tested groups. The total antimicrobial resistance (AMR) load (Figure 9) was significantly higher in the caeca of birds reared on the conventional farm in comparison to the antibiotic-free farm (Wilcoxon rank sum test *p*-value < 0.000) (Figure 9). However, in both groups of caeca samples, the total AMR load was lower in comparison to the carcasses which did not show significant differences between the two tested groups (Wilcoxon rank sum test *p*-value = 0.6).

The drug classes identified in the broiler caeca and on carcasses are listed in Figure 10. The drug class aminoglycoside was more represented on carcasses in comparison to caeca contents and sulfonamide was identified on carcasses but at very low levels in some caeca. On the contrary, macrolide as well as resistance to other drug classes, including bicyclomicyn, lincosamide, fosfomycin, glycopeptide, pleuromutilin and nitrofuran were mostly identified in the caeca. Figure 10 shows that besides differences in the abundance of specific antibiotic resistance genes described above, in qualitative terms the overall resistome of the caeca of animals reared on the antibiotic-free farm overlaps with that of animals reared on the conventional farm, and the same is observed on carcasses. This result can be explained considering that the antibiotic free flock was reared in a farm where antibiotics have been possibly used in the previous flocks, thus supporting the persistence of ARGs in the farm environment over time.

As for bacteria genera and functional genes, the ARGs identified in the caeca also clustered separately for the conventional and antibiotic free-farms (adonis2 *p*-value = 0.00001) (Figure 11), while in the carcasses this difference was lost (adonis2 *p*-value = 0.4278) (Appendix A).

## 4. Discussion

Currently, conventionally raised poultry continues to dominate the EU poultry industry. However, there is an increasing consumer demand for meat obtained in antibiotic-free rearing cycles. Moreover, in January 2022, the new EU regulations of June 2019 [19] and April 2019 [20] will enter in force, further limiting the use of veterinary medical products and medicated feed in animal productions. In this pilot study, we investigated whether the efforts of raising chickens without the use antibiotics would make any difference in the microbiome of poultry meat eaten by consumers.

The results demonstrated three key findings. The first one is a clear separation between the taxonomic, functional and antibiotic resistance genes in the caeca of the birds reared on the conventional and antibiotic-free farms. This result is due to the fact that each poultry farm has an associated ecosystem due to the geographical and specific environmental conditions, to what chickens eat and drink, to the litter type, to the workers, and certainly to the medications they receive or not. That separation was completely lost on carcasses belonging to the two groups, which did not mirror whatever positive or negative impact the farm ecosystem and rearing conditions had on the chicken caeca. As for the caeca, and also for the carcass microbiomes, there are many contributing factors besides the possible cross contamination during the evisceration. Indeed, the ecosystems interacting with the animals during transport and then during each slaughtering step, including the final refrigeration tunnel, all contribute to the final carcass microbiomes.

The second key finding is that the antibiotic free production resulted in statistically significant lower antimicrobial resistance loads in the caeca of chickens in comparison to the conventional production, thus confirming that besides external sources of ARGs, when antimicrobials are not administered to the animals in the caeca of that flock there is a lower antimicrobial resistance load. In relation to the short- and long-term effects of the use of antimicrobials on antimicrobial resistance, Mughini-Gras et al. 2021 [34] showed that the antimicrobial use at flock level is more relevant for antimicrobial resistance in *Escherichia coli* than the historical use of antimicrobials at the farm level. Overall, these observations demonstrate that the effort of reducing antimicrobial use by means of rearing antibiotic free flocks should be associated with a better understanding of the antimicrobial resistance persistence in the farm environment in the absence of direct antimicrobial use. Further insights into the antimicrobial resistance persistence in the farm environment might help us to understand why, for instance, the relative abundance of acriflavine resistance protein genes was higher in the caeca of antibiotic-free animals compared to conventional ones.

The last main finding is that the antimicrobial resistance load on carcasses was much higher than in the caeca, without any significative difference between carcasses coming from the two types of farming. As described above, this result demonstrated that all post-harvest steps, including transport and slaughtering, but also the loading and unloading of the animals contributes not only to the microbiome colonizing the final carcass reaching the consumers, but also to its antimicrobial resistance load. Therefore, although the most important antimicrobial resistance risk factors and possible mitigation measures are still under investigation at the farm level [35], the implementation of past and future EU regulations aimed at reducing antimicrobial use for food production animals has been ensuring a significant reduction of antimicrobial resistance load at the farm level. Therefore, the same effort made for the identification of relevant sources of pathogen and spoilage microorganisms and their ARGs should be now devoted to the post-harvest steps [36]. Little currently available data demonstrate that both transport trucks and cages can contaminate the birds with bacteria and ARGs [37] and contribute to the cross contamination between the slaughterhouse and the farms [38]. Moreover, when the animals reach the slaughtering line, scalding, defeathering and evisceration can spread both microorganisms and ARGs from the animal to the environment, although some tentative steps toward reducing these cross contaminations using innovative technologies are in place [39]. Additional sources of both microorganisms and ARGs are workers, equipment, air, process water and wastewater from slaughtering [40]. All these sources together contribute to the carcass microbiome, and our results showed that at the end of the refrigeration tunnel the microbiome of carcasses from animals reared in the conventional farm overlaps with that of carcasses from birds reared in an antibiotic-free cycle. Our results are consistent with those of Li et al., 2020 [41] who investigated chicken breast microbiomes at the retail level, accounting also for the effect of the processing environment and packaging conditions. Their results confirmed that the microbiome of the chicken breast is affected by packaging in air versus under vacuum and by the processing plant where the chicken breast is processed. On the contrary, both the use of antimicrobials at the farm level as well as seasonality affected neither the composition nor the diversity of chicken breast microbiomes in terms of both alpha and beta diversities.

The alpha diversity calculated in this study at genus level using the indicators of richness (Chao1), evenness (InvSimpson) and diversity (Shannon) within the caeca samples showed values significantly higher in the caeca of birds reared on the conventional farm in comparison to the antibiotic-free farm. The bacteria biodiversity within the GI tract is considered an indicator of good health, and it was expected to be higher in the caeca of chickens not treated with Amoxicillin and Sulfadimethoxine/Trimethoprim. However, these antibiotics are only partially absorbed in the gut [42,43], and this might explain why their administration did not reduce the overall bacteria richness in the caeca. Moreover, the intestinal microbiota biodiversity is the result of different factors, as management protocols applied on the farm, animal characteristics and administered diets [44] which were possibly different in the conventional and antibiotic free farm investigated in this research. Among the most represented genera detected in the caeca, *Alkaliphilus*, *Desulfibacterium*, *Bacillus* and *Ethanoligenens* were identified as signature genera in the birds from the conventional farm, while butyrate-producing microorganisms such as *Coprococcus*, *Roseburia* and *Subdoligranulum* were identified in the caeca of birds reared on the antibiotic-free farm. This result highlighted that besides the higher bacteria biodiversity identified in the caeca of birds from the conventional farm, the signature genera colonizing the caeca of the birds reared on the antibiotic-free farm belonged to microbial groups supporting animal health. Indeed, butyrate fights against pathogen colonization in poultry [45] and is involved in several intestinal functions, being an energy source stimulating epithelial cell proliferation and differentiation, other than exerting an antimicrobial effect by promoting the production of peptides and stimulating the production of tight junction proteins [46].

In this pilot study we investigated for the first time the microbiome of the caeca of a bird and that of the corresponding carcass. The results showed that the caeca and carcasses of the same flock positively correlate one with the other. However, the correlation between the microbiome of the caeca and the carcass of the same bird was not stronger than that with other caeca and carcasses of the same flock. Therefore, the target analysis of caeca and carcass of the same animal does not provide any added value in comparison to the microbiome analysis at flock level. It is also clear from Figure 8 that the correlation between the functional genes was higher than for bacteria genera, possibly because the same functional gene can be shared between different bacteria genera.

Besides the qualitative and quantitative differences in the most represented functional genes identified in the caeca and on carcasses from the animals reared in the conventional and antibiotic free farm, the most relevant result concerns the antibiotic resistant genes and the total antimicrobial resistance load. In relation to the ARGs, the multi antimicrobial extrusion protein (Na(+)/drug antiporter)-MATE family of MDR efflux pumps was significantly higher in the caeca of birds reared on the conventional farm in comparison to the antibiotic-free farm, along with few other ARGs. The MATE gene family is widely distributed in both Gram-positive and Gram-negative bacteria and contributes to the intrinsic, acquired, and phenotypic resistance of bacterial pathogens [47]. Moreover, it can confer resistance to a specific class of antibiotics or to many drugs, thus conferring a multi-drug resistance (MDR) phenotype to bacteria [48]. In contrast, the abundance of genes encoding the acriflavin resistance protein was significantly higher in the caeca of birds reared in the antibiotic free farm. The acriflavine resistance protein is among the multidrug resistance efflux transporter proteins that belongs to the resistance modulation division superfamily (RND), conferring broad spectrum resistance to Gram-negative bacteria [49].

For both caeca and carcasses, the overall antimicrobial resistance abundance per drug class did not show significative difference between the birds collected in the two tested farms, while specific differences were observed between drug classes associated to caeca and carcasses. The drug classes identified in both caeca and carcasses largely overlap with those identified by Munk [50] in the fecal resistome investigated in European poultry farms, including Italian farms. In both studies, aminoglycoside, β-lactam, tetracycline and macrolide are widely represented, although we identified a larger proportion of β-lactam as well as rifamycin not reported by Munk et al. Our results confirmed what was observed by Li et al., 2020 [41] in relation to the absence of difference between the resistome associated to chicken breast from birds reared in conventional and antibiotic-free farms. On the contrary, the results on the antimicrobial resistance load are the opposite because we calculated a higher antimicrobial resistance load on carcasses, while Li et al. discovered a low risk of ARG accumulation on chicken breast. This result is possibly because in the US, poultry carcasses can be disinfected using chlorinated water or organic acids, while in the European Union the use of substances intended to remove microbial surface contamination is only permitted after a full risk analysis taking into account the results of a risk assessment based on the available scientific evidence [51].

Genes coding for resistance to vancomycin were identified among the ARGs with normalized mean values of abundance >1000 in at least one tested group. In accordance with other authors [40,52], vancomycin resistance genes can be identified in poultry flocks, although avoparcin has been banned by the EU since 1997. The relative abundance we estimated for the vancomycin resistance genes constitutes a body of evidence of their persistence, while Savin et al. [40] reported a declining trend.

Overall, the results of this pilot study and the scientific literature demonstrate that each intervention in whatever processing step that the chicken and poultry meat is at, as with other food productions, is affected by the existing microbiome and resistome shifting and changing from farm to fork. Therefore, building robust, comparable, and representative databases of animal-, farm-, food- and production environment-associated microbiomes and resistome from farm to fork, as it is done for individual foodborne isolates and indicator microorganisms [53], would certainly help to predict the effect of control strategies to reduce food contamination by foodborne pathogens as well as ARGs in a systemic way.

## 5. Conclusions

All in all, the results of this research indicate that post-harvest steps withdraw the positive effects of antibiotic-free rearing on carcass microbiomes. Therefore, it is crucial to assess the contribution of both transport and slaughter on carcass contamination and spreading of ARGs to identify possible mitigation options addressing consumer concerns regarding antimicrobial resistance and enhancing the positive impact of the European legislation, as well as the economic and management efforts of producers to rear antibiotic-free chickens.

## Figures and Tables

**Figure 1 foods-11-00249-f001:**
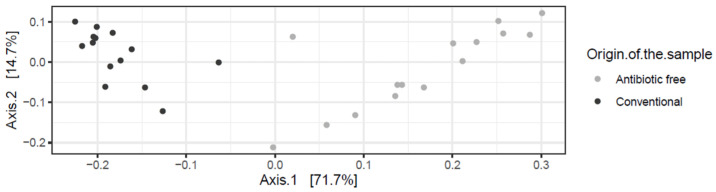
Bray-Curtis dissimilarity plots showing the genera detected in the caeca of birds from the conventional and antibiotic free farm.

**Figure 2 foods-11-00249-f002:**
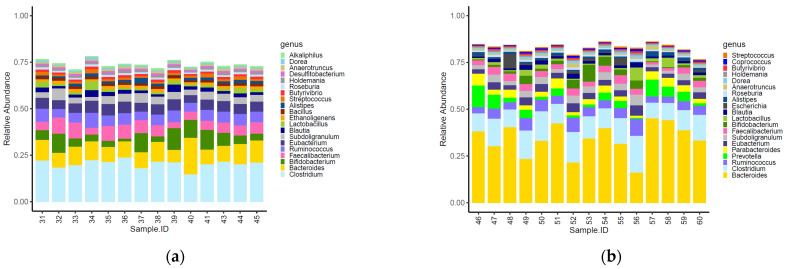
Top 20 genera characterizing the caeca of the birds reared in the conventional (**a**) and antibiotic free (**b**) farm. Sample 42 is not included among samples in panel a because it was not processed for technical reasons.

**Figure 3 foods-11-00249-f003:**
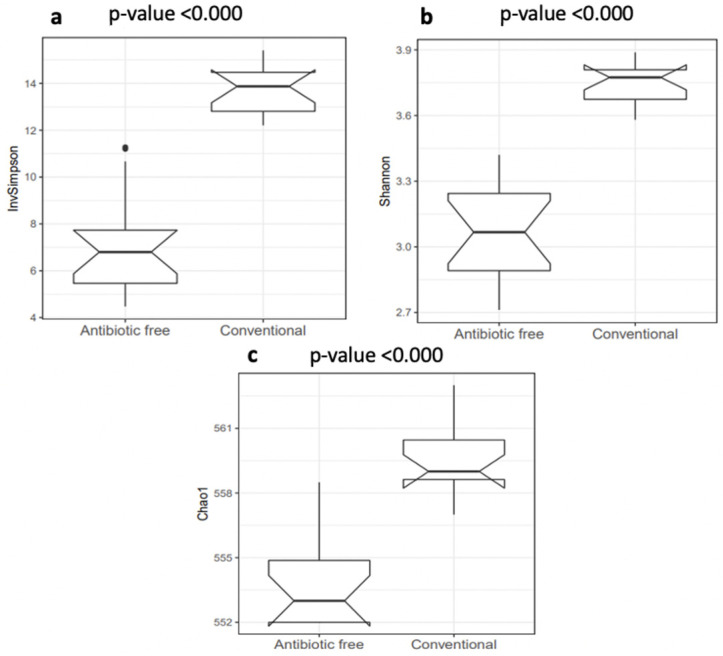
Box plots of the alpha diversity indexes calculated at genus level: (**a**) InvSimpson index; (**b**) Shannon index; (**c**) Chao1 index. *p* values < 0.001 were considered significantly different.

**Figure 4 foods-11-00249-f004:**
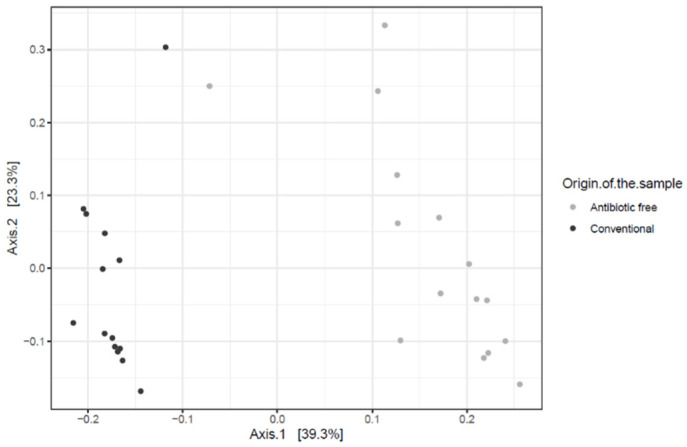
Bray-Curtis dissimilarity plots showing the functional genes detected in caeca of birds reared in the conventional and antibiotic free farm.

**Figure 5 foods-11-00249-f005:**
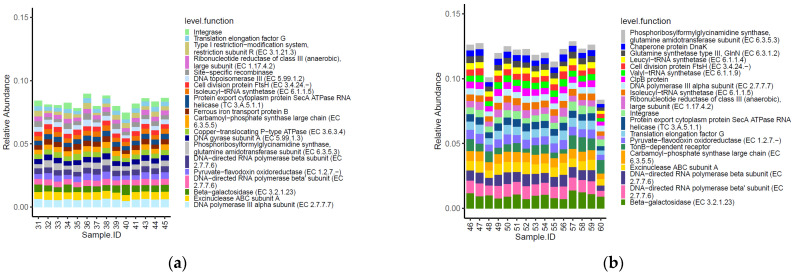
Top 20 functional genes characterizing the caeca of the birds from the conventional (**a**) and antibiotic free (**b**) farm. Sample 42 is not included among samples in panel a because it was not processed for technical reasons.

**Figure 6 foods-11-00249-f006:**
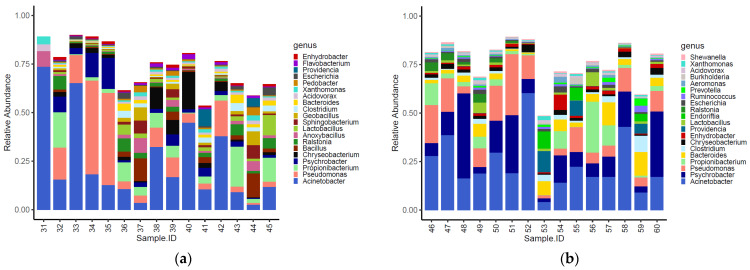
Top 20 genera characterizing the carcasses of the birds reared in the conventional (**a**) and antibiotic-free (**b**) farm.

**Figure 7 foods-11-00249-f007:**
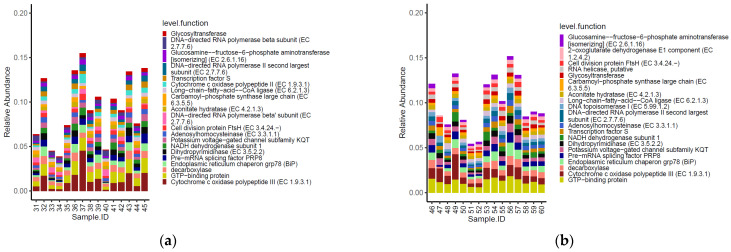
Top 20 functional genes characterizing the carcasses of the birds from the conventional (**a**) and antibiotic free (**b**) farm.

**Figure 8 foods-11-00249-f008:**
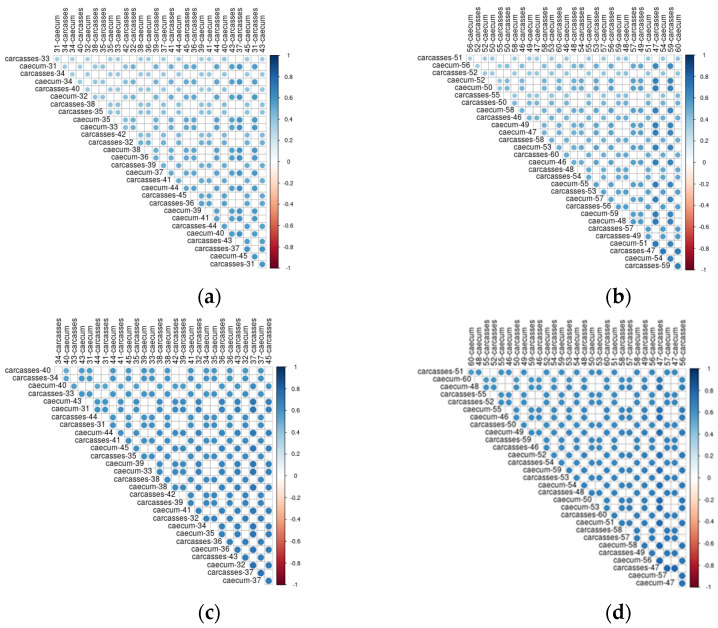
Kendall correlation plot of genera (**a**) and functional genes (**c**) identified in the caeca and corresponding carcasses from the conventional farm and of genera (**b**) and functional genes (**d**) identified in the caeca and corresponding carcasses from the antibiotic–free farm.

**Figure 9 foods-11-00249-f009:**
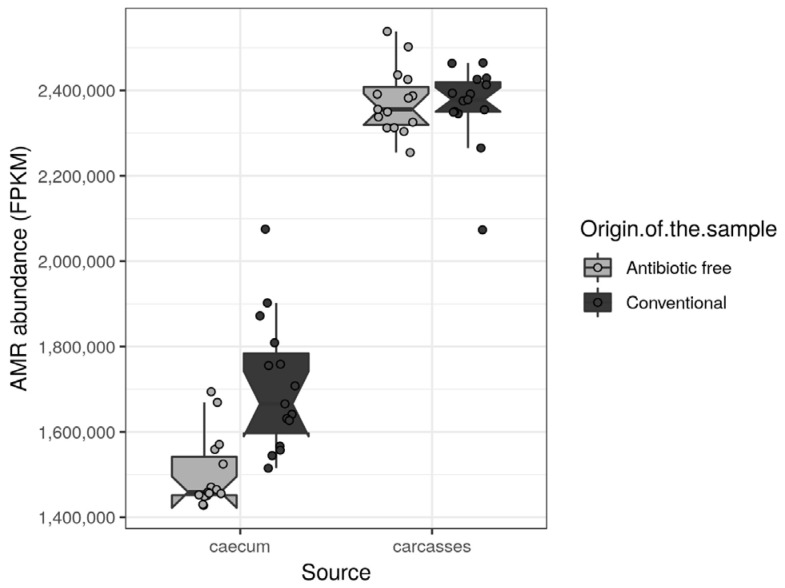
Box plots showing the total AMR level (FPKM) per sample, stratified by source and origin of the sample. Each sample is also represented by a dot with sideways jitter to minimize overplotting.

**Figure 10 foods-11-00249-f010:**
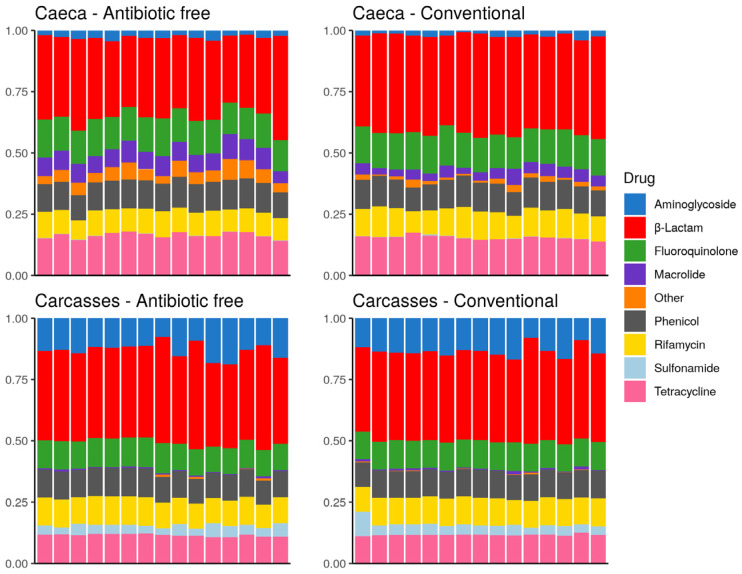
Stacked bar chart of antimicrobial resistance abundance (FPKM) per drug class (colors) per sample (x axis). Each plot refers to one group of samples as detailed in the sub-titles.

**Figure 11 foods-11-00249-f011:**
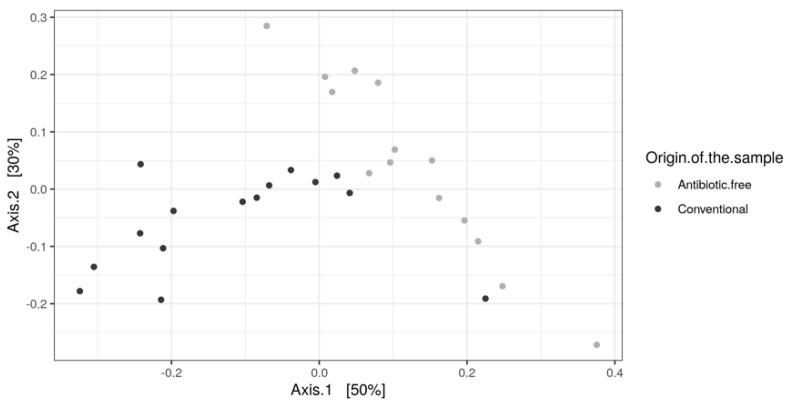
Beta diversity of caeca samples shown as PCoA of Bray–Curtis diversity computed based on the AMR gene family abundances normalized with DESeq2.

## Data Availability

The 59 metagenomes sequenced in this study are publicly available from MG RAST at https://www.mg-rast.org/mgmain.html?mgpage=project&project=mgp89213 (accessed on 14 December 2021).

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
