# Peer review of "Pilot Study on Poultry Meat from Antibiotic Free and Conventional Farms: Can Metagenomics Detect Any Difference?"

_foods, 2022, doi:10.3390/foods11030249_

Round 1
Reviewer 1 Report
None
Reviewer 2 Report
I am happy to see that the authors have made all significant and relevant changes from previous revisions, I really appreciate it. The manuscript is suitable to be accepted in the present form.
Reviewer 3 Report
Dear authors
Thank you for making the suggested changes
This manuscript is a resubmission of an earlier submission. The following is a list of the peer review reports and author responses from that submission.
Round 1
Reviewer 1 Report
I greatly appreciated your manuscript. This is interesting for both scientific community and the consumer . The antibiotic-free management without an optimisation of the biosecurity levels to all segments of chain production has poor meaning . However I suggest adding age and sex of birds at slaughter . The acriflavin resistance protein higher in AF farm is also an interesting result for possible speculations . Finally the manuscript is mostly well written, only the discussion should be "reconsidered" in relation to the use of English language
Reviewer 2 Report
De Cesare and colleagues compared caeca and carcasses of broilers that received or not an antimicrobial treatment. They found a connection with gut microbiota but not with bacteria on the carcasses due to slaughter contamination. The sample size is limited to a single flock. The major issue of this study is the lack of a Time-zero sampling in order to demonstrate that before antimicrobial treatments the bacterial populations and antimicrobial resistance were similar in the two groups. In order to obtain this baseline similarity animals should come from the same hatchery, the same breeders and the effect of environmental contamination should be excluded (eg strict procedures of cleaning and disinfection, ect.).
Title: title can be misleading. It was just one flock from one farm. We also do not know whether this farm is routinely or occasionally antibiotic free. There are short-term and long-term effects of antimicrobial usage on antimicrobial resistance, that should be taken into account and discussed.
Line 34: you should add a ref proving the very low AMU in broilers according to dose-based methods
Line 83-84: the second part of the sentence seems to me a bit unclear, please rephrase.
Lines 89-98: please include for both groups: animal genetics, gender, live weight, duration of the cycle. The total amount in mg/kg of Amoxicillin and Sulfadimethoxine should be provided, and possibly the total DDDvet or DDDita per cycle. Please also include the reasons behind the treatment and the choice of those antibiotics. Clarify the management in antibiotic free farms and procedures applied when antimicrobials are needed.
Line 289: you should discuss resistance to vancomicine, that is not authorised in veterinary medicine.
Reviewer 3 Report
In this manuscript by “Cesare et al”, authors performed a pilot study to compare the microbiomes of two groups of chickens: one group of chickens reared in conventional and another group of chickens reared in antibiotic free farms. Overall, the manuscript is well written and presented. However, I have minor comments:
1. The author needs to discuss about the impact to the feed. Does the feed supplemented was free of antimicrobials to the birds reared in the antibiotic free farm.
2. Line 431 - The authors have observed that acriflavin resistance protein was significantly higher in the caeca of birds reared in the antibiotic free farm. Any specific reason for that?
3. Line 440 - The authors have stated that the overall antimicrobial resistance abundance per drug class did not show significative difference between the birds collected in the two tested farms, what can be inferred from that?
Reviewer 4 Report
In the paper “Poultry meat from antibiotic-free and conventional farms: Is there a difference?”, the authors report on the differences in microbiomes and functional genes present in the caeca and corresponding carcasses on broilers reared in conventional and antibiotic free farms. This is a timely study considering the expected changes that will occur when EU Regulation 2019/6 and 2019/4 comes into effect in 2022. Overall, a well written and designed study with a few minor editorial changes and major comments to address.
Title – The title should be changed to reflect the fact that the study is looking at differences in the microbiome of poultry meat rather than differences in texture or sensory characteristics.
Introduction
Line 58-59 - change to “resistance is responsible for 33,000 death per year”
Line 59 - change to “the use of antibiotics”
Line 75 – change to “the wide use of vaccination”
Materials and Methods
Line 93 – Do we know what the concentrations were of these antibiotics and how they were administered?
Line 101 – change to “second refrigerated box”
Line 108 – change the centrifugal parameters from rpm to x g. Also what was the purpose of centrifuging the homogenised sample at this step?
Line 116 – The authors state that DNA was extracted from the carcass as previously described and provide a reference (reference 23). However, following this reference doesn’t provide any further information that doesn’t already appear in the text. In contrast, reference 22 goes into greater detail on the bead beating procedure. I suggest reference 23 should be removed from the paper.
Results
Figure 2 – Is it possible to change the colour scheme of a) and b) so they are matching e.g. Clostridium is sky blue in figure a but orange in figure b. It would allow observing and comparing the two groups much easier.
Figure 5 – same as above and with all other diagrams.
Also Figure 2 and 5 – a technical note should be included in these diagrams explaining why sample 42 isn’t included in this analysis (explained in line 121 – 123 but nevertheless should also be included in the description of these diagrams).
Figure 2 – is it possible that antibiotics used in this study may potentially select for probiotic bacteria? Both Lactobacillus and Bifidobacterium in the chicken caeca have demonstrated benefits to the chicken immune system yet their populations are significantly reduced in the antibiotic-free samples set. Perhaps something to touch upon in the discussion section.
I’m not sure if analysing the functional genes contributes anything to this study. The genes reported are involved in cell division, transport and DNA replication which is to be expected. The significance of these results are also barely touched upon in the discussion/conclusion.
Reviewer 5 Report
Dear authors,
What does secondo mean (Line 100, Page 3)
The presented work is interesting and the writing clear. However the title does not allow to see what is the focus of the research. My suggestion is to be more explicit